# SepF is the FtsZ anchor in archaea, with features of an ancestral cell division system

Nika Pende[1,8], Adrià Sogues[2,8], Daniela Megrian[1,3], Anna Sartori-Rupp[4], Patrick England [5], Hayk Palabikyan[6], Simon K.-M. R. Rittmann [6], Martín Graña [7], Anne Marie Wehenkel [2✉], Pedro M. Alzari [2] & Simonetta Gribaldo [1✉]

Most archaea divide by binary fission using an FtsZ-based system similar to that of bacteria, but they lack many of the divisome components described in model bacterial organisms. Notably, among the multiple factors that tether FtsZ to the membrane during bacterial cell constriction, archaea only possess SepF-like homologs. Here, we combine structural, cellular, and evolutionary analyses to demonstrate that SepF is the FtsZ anchor in the human-associated archaeon *Methanobrevibacter smithii*. 3D super-resolution microscopy and quantitative analysis of immunolabeled cells show that SepF transiently co-localizes with FtsZ at the septum and possibly primes the future division plane. *M. smithii* SepF binds to membranes and to FtsZ, inducing filament bundling. High-resolution crystal structures of archaeal SepF alone and in complex with the FtsZ C-terminal domain (FtsZ$_{CTD}$) reveal that SepF forms a dimer with a homodimerization interface driving a binding mode that is different from that previously reported in bacteria. Phylogenetic analyses of SepF and FtsZ from bacteria and archaea indicate that the two proteins may date back to the Last Universal Common Ancestor (LUCA), and we speculate that the archaeal mode of SepF/FtsZ interaction might reflect an ancestral feature. Our results provide insights into the mechanisms of archaeal cell division and pave the way for a better understanding of the processes underlying the divide between the two prokaryotic domains.

[1] Evolutionary Biology of the Microbial Cell Unit, CNRS UMR2001, Department of Microbiology, Institut Pasteur, Paris, France. [2] Structural Microbiology Unit, Institut Pasteur, CNRS UMR 3528, Université de Paris, Paris, France. [3] École Doctorale Complexité du vivant, Sorbonne University, Paris, France. [4] Ultrastructural BioImaging Unit, Institut Pasteur, Paris, France. [5] Plate-forme de biophysique moléculaire, C2RT-Institut Pasteur, CNRS, UMR 3528, Paris, France. [6] Archaea Physiology & Biotechnology Group, Department of Functional and Evolutionary Ecology, University of Vienna, Wien, Austria. [7] Bioinformatics Unit, Institut Pasteur of Montevideo, Montevideo, Uruguay. [8] These authors contributed equally: Nika Pende, Adrià Sogues. ✉email: anne-marie.wehenkel@pasteur.fr; simonetta.gribaldo@pasteur.fr

Traditionally viewed as inhabitants of extreme environments, the Archaea are now fully recognized as ubiquitous prokaryotes of great ecological and evolutionary importance[1–3]. Additionally, the human "archaeome"—whose research is still in its infancy—is receiving growing attention, as an imbalance of archaeal methanogens has been linked to various pathologies such as Inflammatory Bowel Disease, multiple sclerosis, anorexia and colorectal cancer[4]. Despite this potential clinical relevance, knowledge on the process of cytokinesis and the involved actors is still very partial in Archaea with respect to Bacteria. While some Archaea (Crenarchaeota) divide by using homologs of the eukaryotic ESCRT system[5,6], most archaeal genomes possess one or two homologs of FtsZ (FtsZ1 and FtsZ2)[7], and some also other tubulin-like proteins such as the CetZ family[8,9]. The role of FtsZ during cytokinesis has been shown in a few model archaeal organisms[10–13], such as *Haloferax volcanii* where FtsZ1 localization at the mid-cell constriction site provided the first cytological evidence of an archaeal FtsZ role in cell division[13]. A more recent study showed that *H. volcanii* FtsZ1 and FtsZ2 co-localize at midcell as a dynamic division ring, with FtsZ1 involved in stabilization of the FtsZ2 ring and in cell shape, while FtsZ2 is involved in cell constriction[10].

Only few homologs of the bacterial division machinery that interact with FtsZ have been identified in Archaea, such as the positive and negative FtsZ regulators SepF and MinD, respectively[7,14,15]. SepF was originally identified as a component of the *Bacillus subtilis* divisome required for correct septal morphology and that interacts with the C-terminal domain of FtsZ (FtsZ$_{CTD}$)[16,17]. In *B. subtilis*, SepF is non-essential and has an overlapping role with FtsA, as overexpression of SepF can rescue an FtsA depletion strain[16,17]. In contrast, in other bacterial lineages where FtsA is absent, such as most Actinobacteria and Cyanobacteria, SepF is an essential component of the early divisome, orchestrating Z-ring assembly and participating in membrane remodeling[18–21]. The first crystal structure of a bacterial SepF-FtsZ complex was only recently obtained from the Actinobacterium *Corynebacterium glutamicum*[21]. It showed that SepF forms a functional dimer that is required for FtsZ binding[21]. *C. glutamicum* SepF binds to the FtsZ$_{CTD}$ through a conserved pocket and interacts with residues at the α-helical interface of the functional dimer[21].

An early analysis of the distribution of cell division proteins in Archaea revealed that SepF is present in almost all FtsZ-containing taxa, which led to the suggestion that it could act as the main FtsZ anchor[7]. The structures of two archaeal SepF-like proteins have been reported[22], but their biological function has not been studied.

Here we provide experimental evidence for a functional role of archaeal SepF in cytokinesis through interaction with FtsZ by using as model *Methanobrevibacter smithii*, the most abundant species of archaeal methanogens in the human microbiome[4]. *M. smithii* contains only one copy of FtsZ1 (*Ms*FtsZ) and one copy of SepF (*Ms*SepF) and is therefore an interesting model with respect to the more complex Haloarchaea which have two FtsZ copies in addition to up to six tubulin-like proteins of the CetZ family[8]. As no genetic tools are currently available for *M. smithii*, we used an integrative study combining cell imaging, protein biochemistry and structural analysis. We demonstrate that SepF co-localizes with FtsZ to midcell during cell constriction and binds to membranes. Structural and biochemical analysis shows that SepF also forms a functional dimer which binds the FtsZ$_{CTD}$ via a pocket that is partially conserved with Bacteria, but with a markedly different interaction pattern occurring through the β−β interface. Finally, we complement these results by a thorough evolutionary analysis of SepF and FtsZ in Bacteria and Archaea. This reveals that SepF and FtsZ were already present in the Last Universal Common Ancestor, and that the Archaea might have retained features of an ancestral minimal cytokinesis machinery, while Bacteria diverged substantially, likely to accommodate the emergence of a rigid cell wall and the complex divisome.

## Results

**SepF transiently co-localizes with FtsZ during the *M. smithii* cell cycle**. To investigate the function of *Ms*SepF and its potential interaction with *Ms*FtsZ, we studied the localization pattern of these two proteins during the *M. smithii* cell cycle by immunolabelling, using specific anti-*Ms*FtsZ and anti-*Ms*SepF antibodies (Methods, Fig. 1). The cell wall of *M. smithii* (as well as all Methanobacteriales and the sister order Methanopyrales) consists of pseudo-peptidoglycan (pPG)[23,24]. The archaeal pPG oligosaccharide backbone is composed of L-N-acetyltalosaminuronic acid with a β−1,3 linkage to N-acetylglucosamine and the stem peptide is made of only L-amino acids[23]. These structural differences compared to bacterial PG make these archaea resistant to most lysozymes and proteases[23]. Therefore, we developed a new protocol for immunolabelling where cells are pre-treated with the phage endoisopeptidase PeiW[25,26] which specifically permeabilizes the pPG cell wall (Methods). Using Western Blot analysis, we quantified FtsZ and SepF levels and showed that the FtsZ/SepF molar ratio is about six during exponential growth (Supplementary Fig. 1). To study the intracellular localization of *Ms*FtsZ and *Ms*SepF, we performed co-immunolabelling with anti-*Ms*SepF and anti-*Ms*FtsZ antibodies and imaged the cells by 3-dimensional (3D) super-resolution microscopy (Methods, Fig. 1a). In the ovococcoid *M. smithii* cells, *Ms*FtsZ forms a patchy ring-like structure at mid-cell and *Ms*SepF largely overlaps with it (Fig. 1a, left panel, Supplementary Movie 1). At a later stage of division, a smaller FtsZ ring corresponds with the almost completed cell constriction and two new rings appear in the future daughter cells, defining the new septation planes, with SepF forming discontinuous arcs largely overlapping with FtsZ (Fig. 1a right panel, Supplementary Movie 2).

We then performed quantitative analysis with several hundreds of co-immunolabeled *M. smithii* cells imaged by conventional epifluorescence microscopy (Methods, Fig. 2b and c). Our results show that in cells exhibiting very slight or no constriction, *Ms*SepF and *Ms*FtsZ co-localize at the future septation plane (Fig. 1b, cell 1; 2c(i); Supplementary Fig. 2). In cells that start to constrict, *Ms*FtsZ is present at the septation plane and *Ms*SepF is found slightly lateral of the Z-ring (Fig. 2b, cells 2–3; 2c(i); Supplementary Fig. 2). At a later stage, two distinct fluorescent *Ms*SepF foci can be seen further away from the single *Ms*FtsZ focus, suggesting that SepF moves to the future division planes of the prospective daughter cells before FtsZ (Fig. 2b, cells 4–5; 2c (ii); Supplementary Fig. 2). Finally, in strongly constricted cells, FtsZ appears in the prospective daughter cells overlapping with the SepF signal, but it is also still present in the current septation plane, where only a weak SepF signal can be detected (Fig. 2b, cell 6; 2c(iii)).

From these results, we propose a model for FtsZ and SepF localization during the *M. smithii* cell cycle (Fig. 1d): (i) in non-constricting cells, both proteins co-localize at the septation plane; (ii and iii) as constriction starts and proceeds, SepF primes the future septation plane in the prospective daughter cells by progressively moving there before FtsZ; (iv) in cells that have almost completed constriction, FtsZ and SepF co-localize at the prospective septation plane of the daughter cells.

**$\textit{Ms}$SepF binds to membranes and to the FtsZ$_{CTD}$ inducing filament bundling**. *Ms*SepF is composed of 149 amino acids and presents the same conserved domain organization as previously described for bacterial SepF proteins[21,22] consisting of an N-terminal amphipathic helix representing the putative membrane-

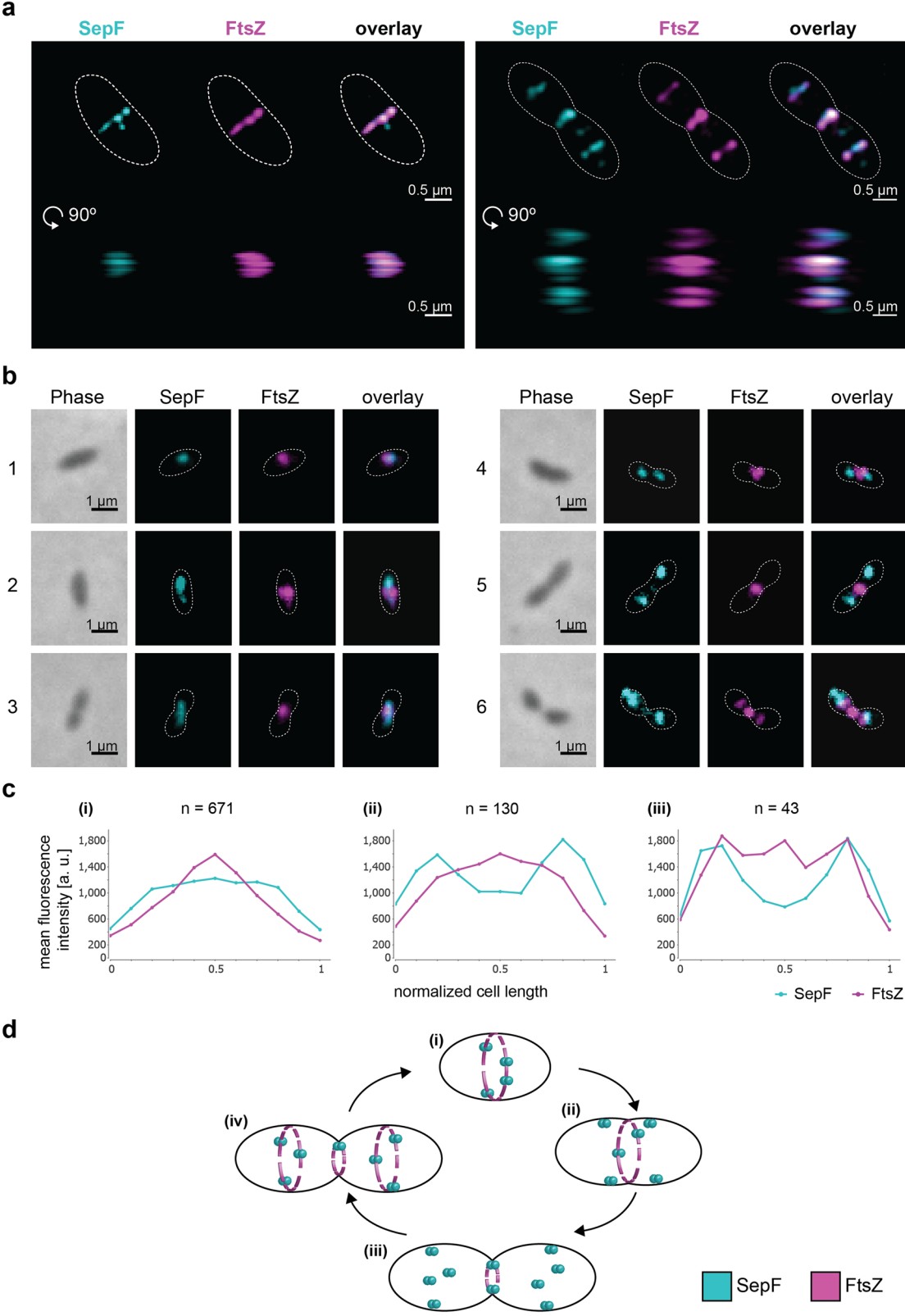

binding domain (residues 1-12) connected through a flexible linker to a putative C-terminal FtsZ-binding core (residues 54 to 149, Fig. 2a, Supplementary Fig. 3). We show that the membrane-binding domain of *Ms*SepF interacts with small unilamellar vesicles (SUVs) with an affinity of 43 ± 0.2 μM (Methods, Supplementary Fig. 4a), a value in the same range as that reported for *C. glutamicum*[21]. Moreover, thermal shift assays show an

important increase of the melting temperature of *Ms*SepF$_{core}$ (from 70.6 to 82.3 °C) upon addition of *Ms*FtsZ$_{CTD}$. Supplementary Fig. 5a), indicating a direct interaction between the two proteins. The apparent Kd value for this interaction is 84.0 ± 8.5 μM as determined by surfaces plasmon resonance (SPR) (Supplementary Fig. 5b), a value in the same range as the one described for *C. glutamicum*[21].

**Fig. 1 SepF co-localizes with FtsZ during the *M. smithii* cell cycle.** *M. smithii* cells were permeabilized with PeiW and immunostained with anti-*Ms*SepF and anti-*Ms*FtsZ antibodies. **a** 3D Structured Illumination Microscopy (SIM) maximum projections of a non-constricting *M. smithii* cell (left panel) and a constricting cell (right panel) stained with anti-*Ms*SepF (cyan) and anti-*Ms*FtsZ antibodies (magenta). Front views of single channels and the overlay are depicted (above) as well as the side view shifted by 90° (below). White dotted lines represent the cell outlines and scale bars are 0.5 μm. **b** Phase contrast (Phase) and corresponding epifluorescence images of representative co-labeled *M. smithii* cells (SepF in cyan, FtsZ in magenta and an overlay of both). Cells are arranged from non-constricting to constricting from 1 to 6. White dotted lines represent the cell outlines deduced from the corresponding phase contrast images and the scale bars are 1 μm. **c** Mean fluorescence intensity plots of cells grouped into three classes according to the detected FtsZ fluorescent maxima (0–1, 2 or 3 maxima detected) with the corresponding SepF (cyan) and FtsZ (magenta) mean fluorescence intensity [a. u.] of each group plotted against the normalized cell length [0-1]. **d** Schematic view of SepF (cyan) and FtsZ (magenta) localization pattern during the life cycle of *M. smithii*. (i) In non-constricting cells, both proteins co-localize at the septation plane. (ii) and (iii) SepF progressively moves to the future division plane prior FtsZ. (iv) As constriction is almost completed, FtsZ and SepF co-localize at the prospective septation plane of the daughter cells. The data shown here are representative for experiments performed in triplicate.

### Table 1 Crystallographic data.

|  | **MsSepF$_c$** | **MsSepF$_c$ + FtsZ$_{CTD}$** |
|---|---|---|
| ***Data collection*** |  |  |
| Space group | P 3$_1$ 2 1 | P 6$_1$ 2 2 |
| Cell dimensions |  |  |
| a, b, c (Å) | 53.05, 53.05, 53.91 | 64.85, 64.85, 107.92 |
| α, β, γ (°) | 90, 90, 120 | 90, 90, 120 |
| Resolution (Å)* | 45.95–1.4 (1.42-1.4) | 38.91 − 2.7 (2.83 − 2.7) |
| R$_{sym}$ | 0.056 (1.029) | 0.076 (0.702) |
| I/σ(I) | 27.1 (3.4) | 15.4 (2.7) |
| Completeness (%) | 99.9 (100) | 95.7 (96.3) |
| Redundancy | 19.6 (19.9) | 5.5 (5.7) |
| CC ½ | 0.999 (0.921) | 0.999 (0.918) |
| ***Refinement*** |  |  |
| Resolution (Å) | 1.4 | 2.7 |
| Number of reflections | 17673 | 3806 |
| R-work/R-free | 0.182 / 0.216 | 0.229 / 0.255 |
| Number of atoms |  |  |
| protein | 684 | 746 |
| ligands/ions | 8 | — |
| water | 57 | — |
| B-factors (Å$^2$) |  |  |
| protein | 25.47 | 61.5 |
| ligands/ions | 25.49 | — |
| water | 37.47 | — |
| RMS deviations |  |  |
| Bond length (Å) | 0.009 | 0.008 |
| Bond angles (°) | 1.057 | 1.02 |
| ***PDB code*** | 7AL1 | 7AL2 |

* Values in parenthesis refer to the highest recorded resolution shell.

As shown for *C. glutamicum* SepF (*Cg*SepF)[21], or *B. subtilis*[20], *Ms*SepF is also able to remodel SUVs, as observed by negative stain EM (Fig. 2b(i)), and this remodeling was reversed upon addition of the FtsZ$_{CTD}$, giving rise to smaller, more regular lipid vesicles (Fig. 2b(ii), the control of SUVs alone is shown in Supplementary Fig. 4c). Moreover, we confirmed that the *Ms*SepF$_{core}$, which is lacking the MTS and linker domains does not polymerize in the presence of SUVs (Supplementary Fig. 4b). Additionally, *Ms*SepF$_{core}$ is capable of inducing bundling of FtsZ protofilaments, giving rise to longer and straighter filaments than FtsZ-GTP alone (Fig. 2c and d). Taken together, these results strongly suggest conservation of common functional features between archaeal and bacterial SepF.

**High-resolution crystal structures of *Ms*SepF$_{core}$ alone and in complex with the FtsZ$_{CTD}$, reveal specific archaeal features.** We determined the crystal structure of *Ms*SepF$_{core}$, at 1.4 Å resolution

(Fig. 3a, Table 1). The protein is a dimer in solution (Supplementary Fig. 6), with each protomer consisting of a 5-stranded β-sheet flanked by 2 α-helices (Fig. 3a). In addition, a helical turn insertion (η1), between strand β2 and helix α2, is conserved in archaeal sequences but is absent in bacterial homologs (Supplementary Fig. 7). The overall dimer organization of *Ms*SepF$_{core}$ matches that previously described for the structures of archaeal SepF-like proteins from *Archaeoglobus fulgidus* and *Pyrococcus furiosus*[22] (Supplementary Fig. 8), with the dimer interface (~1200 Å$^2$) formed by the exposed faces of the two central β-sheets, in which the last strand (β5) forms part of the opposing β-sheet (Fig. 3a). Despite their quite similar monomeric structure, the dimeric arrangement of archaeal SepF differs from that observed for the functional dimer in *Cg*SepF, where the dimer interface is formed by a 4 α-helical bundle (Fig. 3b). Interestingly, both interfaces—formed by the α-helices or the β-sheets—have been simultaneously observed in crystals of *Bacillus subtilis* SepF, generating linear polymers of tightly packed SepF dimers[22].

To further characterize the interaction between SepF and FtsZ, we crystallized *Ms*SepF$_{core}$ in complex with the FtsZ$_{CTD}$ peptide and determined the structure of the complex at 2.7 Å resolution (Table 1). The bound peptide is well-defined in the electron density for 10 out of 13 residues (Supplementary Fig. 9) and adopts an extended conformation within a pronounced groove of the SepF monomer (Fig. 4a). The association is mediated by both hydrophobic and electrostatic interactions (Fig. 4b and Supplementary Fig. 10), the latter including 3 intermolecular ion pairs (FtsZ Asp 370 – SepF Arg 130, Asp 371 – Lys 115 and Asp 374 – Arg 118) and seven additional hydrogen bonds. Three of these H bonds are made by backbone atoms of the FtsZ$_{CTD}$ peptide with the strand β3, partially extending the β-sheet (Supplementary Fig. 11a). The archaeal-specific η1 insertion (residues 97-102) promotes the formation of a wider binding pocket compared to bacterial SepF (Fig. 4c) and is involved in a network of hydrogen bonding interactions that contributes to stabilize the FtsZ-SepF complex (Supplementary Fig. 11b).

The residues defining the binding pocket are highly conserved in archaeal SepF homologs (Supplementary Fig. 12), suggesting a similar FtsZ-binding mode. Furthermore, despite their evolutionary distance and sequence divergence, the structures of bacterial and archaeal SepF-FtsZ complexes share a partially conserved mode of interaction for the N-terminal part of the FtsZ$_{CTD}$ (residues LDDFI in *Ms*SepF and DLDV in *Cg*SepF), which binds to a similar groove formed between α2 and β3 in the SepF monomer (Fig. 4c). However, the binding modes of the C-terminal half of FtsZ$_{CTD}$ are markedly different between Archaea and Bacteria. In fact, in *C. glutamicum* this region interacts with residues from the two monomers of the SepF functional dimer at the α-helical interface (Fig. 4c, right panel), while in *M. smithii* it largely binds to only one monomer of the functional dimer

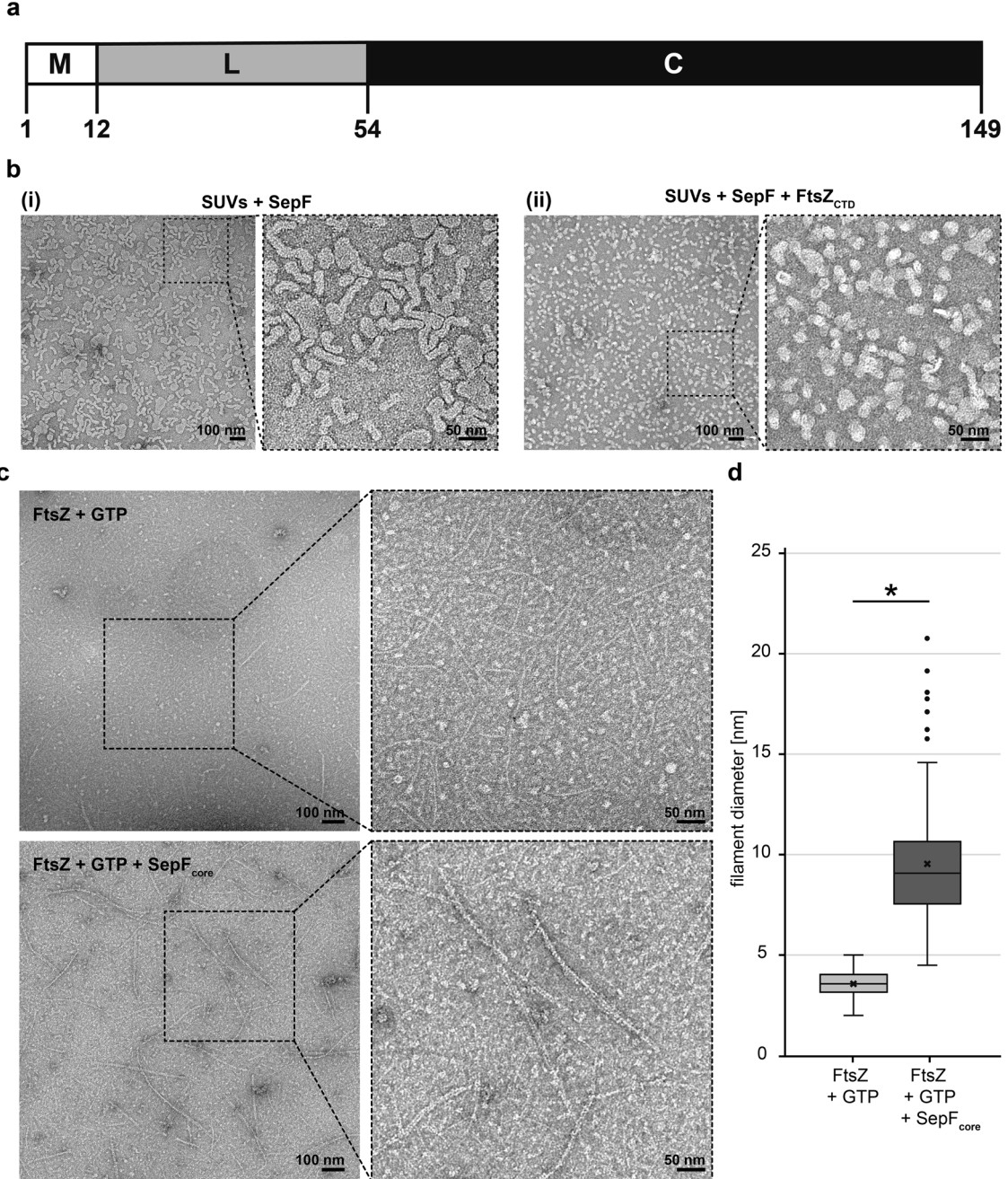

**Fig. 2 *Ms*SepF binds to membranes and to FtsZ_CTD inducing filament bundling. a** Domain organization of SepF from *M. smithii* with an N-terminal amphipathic helix (M), flexible linker (L) and putative C-terminal FtsZ-binding core (C). **b** Negative stain electron microscope images of SUVs (100 μmol L$^{-1}$) and *Ms*SepF (50 μmol L$^{-1}$) with (i) or without (ii) FtsZ_CTD (100 μmol L$^{-1}$). **c** Negative stain electron microscope images of FtsZ (30 μmol L$^{-1}$) and GTP (3 mmol L$^{-1}$) with (upper panel) or without (lower panel) *Ms*SepF_core (20 μmol L$^{-1}$). **b** and **c** Panels show original image with an enlarged inlet that is marked by black dotted square next to it. Scale bars are 100 nm (original images) and 50 nm (inlets). These experiments were performed two times. **d** Boxplots showing filament diameter measurements [nm] for FtsZ + GTP ($n = 130$) and FtsZ + GTP + SepF_core ($n = 130$). Box is the inter quartile range, where the lower edge is 25th percentile and the upper edge the 75th percentile. Whiskers show the range between the lowest value and the highest value. Line inside each box indicates the median and x indicates the mean. Black circles are outliers. A 2-sample t-test was conducted, the result was found to be −23.8032 and the test resulted in a critical t-value of t(alpha) of 1.9692 for an alpha of 0.025. A significant difference of the diameter was found at a 5% level of significance, because the H$_0$ was rejected, as the modulus of the critical value was > α/2. (*) indicates that means of diameter are significantly different.

(Fig. 4c, left panel), enlarging the monomer-peptide interface surface (600 Å$^2$) with respect to that observed in *C. glutamicum* (390 Å$^2$). In *M. smithii*, only the terminal residue of FtsZ_CTD (Phe377) binds to a surface hydrophobic pocket in the second SepF monomer at the β-sheet dimer interface. Phe377 does not seem conserved in other archaeal SepF homologs except for those

phylogenetically close to *M. smithii*. Therefore, the hydrophobic interaction via the FtsZ_CTD terminal residue is likely specific to these taxa.

Archaeal SepF homologs are expected to bind both FtsZ1 and FtsZ2 isoforms, as the conservation pattern of the CTDs from these two isoforms, centered around a nearly invariant GID motif

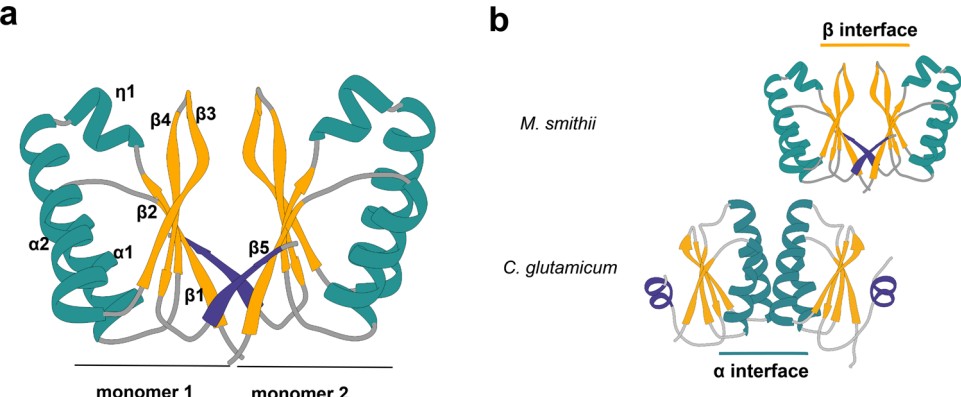

**Fig. 3 Structural characterization of *Ms*SepF_core. a** Crystal structure of the *Ms*SepF dimer composed of two identical monomers, color-coded according to secondary structure (helices, green; strands, yellow). Each protomer consists of a 5-stranded β-sheet flanked by 2 α-helices and a helical turn (η1). In each protomer, the C-terminal strand β5 (in purple) forms part of the opposing β-sheet. **b** Comparison of functional SepF dimer interfaces in Bacteria and Archaea, as defined by the complexes with FtsZ. The α interface has only been found in the crystal structures of bacterial SepF dimers such as those of *C. glutamicum* (PDB 6SCP, shown in the figure) and *B. subtilis* (PDB 3ZIH), whereas the β interface has been found in all Archaeal structures (e.g. *Archaeoglobus fulgidus* (PDB: 3ZIE), *Pyrococcus furiosus* (PDB: 3ZIG), see also Supplementary Fig. 8). The C-terminal secondary structure element of the crystal structures (β5 in *M. smithii* and α3 in *C. glutamicum*) are depicted in purple.

(Fig. 4d, left panel), are very similar to each other. In contrast, the archaeal motif is clearly different from that of the bacterial FtsZ_CTD (Fig. 4d, right panel). In particular, it is interesting to note that the bifurcation pointing the FtsZ_CTD towards the β interface in Archaea is found between the highly conserved isoleucine and aspartate residues of the GID motif (Fig. 4c and d, left panel), further strengthening the argument that this dimer interface was selected early on in archaeal evolution. In contrast, the almost universally conserved proline residue in bacterial FtsZ_CTD is at the bifurcation pointing towards the alpha interface in bacteria (Fig. 4c and d, right panel). Therefore, the different FtsZ-binding modes of *Ms*SepF and *Cg*SepF are driven by their specific self-association (dimerization) modes.

**Distribution and phylogeny of SepF and FtsZ indicates an ancient origin prior to the divergence between Bacteria and Archaea.** The specificities of archaeal SepF/FtsZ interaction with respect to those observed in Bacteria are intriguing and may lie in an ancient and fundamental divergence in cell division between the two prokaryotic domains. We therefore sought to investigate when these two proteins emerged and how they evolved in the two prokaryotic domains.

We updated the distribution of FtsZ1, FtsZ2 and SepF homologs, as well as the presence of the ESCRT-like system in the large number of archaeal genomes that have recently become available[1–3,7,27] (Methods, Fig. 5a, Supplementary Data 1). Most Archaea have one SepF homolog which systematically co-occurs with one or two copies of FtsZ (FtsZ1 and FtsZ2), while it is absent from the genomes that do not have FtsZ homologs. Interestingly, while the genes coding for FtsZ and SepF frequently exist in close proximity in bacterial genomes, this is rarely the case in Archaea, where they co-occur only in a few members of the DPANN superphylum, while they localize with different genes in the other Archaea (*ftsZ1* mostly with components of the translation machinery, *ftsZ2* frequently with *parD*, and *sepF* displaying a more variable genomic context) (Supplementary Fig. 13). Methanopyri are the only Archaea to possess an FtsA-like homolog along with SepF, which likely originated via horizontal gene transfer from bacteria, and whose function is unknown.

In contrast with the wide distribution of SepF in Archaea, its presence in Bacteria is much more scattered, being found mostly in a few phyla belonging to the Terrabacteria clade

(Cyanobacteria and related uncultured candidate phyla, Actinobacteria, Firmicutes, Armatimonadetes, Abtidibacteriota, and the uncultured candidate phylum Eremiobacteraeota), frequently together with FtsA, as opposed to the Gracilicutes (including *Escherichia coli*) which have FtsA only (Fig. 5b and Supplementary Data 2).

The phylogeny of SepF clearly shows a separation of archaeal and bacterial homologs and—although not fully resolved due to the small number of informative positions—an overall topology consistent with vertical inheritance, i.e., monophyly of major phyla and relationships roughly congruent with the reference bacterial and archaeal phylogeny (Supplementary Fig. 14). A similar evolutionary history can be inferred from FtsZ, which shows a clear separation among archaeal and bacterial homologs (Supplementary Fig. 15), and suggests an ancient gene duplication in Archaea giving rise to FtsZ1 and FtsZ2 paralogues, consistent with recent reports[10]. These results strongly suggest that SepF and FtsZ emerged before the divergence of Archaea and Bacteria, likely being part of the division machinery in the Last Universal Common Ancestor (LUCA) and subsequently co-evolved in the two prokaryotic domains. However, while SepF was largely retained in the Archaea, it was lost massively during bacterial diversification, possibly after the emergence of FtsA whose phylogeny shows an early origin in Bacteria (Supplementary Fig. 16).

## Discussion

Cell division is one of the most ancient and fundamental processes of life. Yet, Bacteria and Archaea present profound differences in the way they divide, illustrated by the fact that the majority of components of the bacterial divisome are absent in Archaea. Moreover, in contrast to the well-studied FtsA, the function of SepF has been analyzed in a few bacteria so far[16,17,19–22,28], and only one crystal structure in complex with FtsZ is currently available from *C. glutamicum*[21].

Here, we report the first experimental evidence that archaeal SepF also acts as the FtsZ membrane anchor during cytokinesis. By applying our newly developed immunolabelling protocol for archaeal methanogens with a cell wall made of pPG and 3D super resolution microscopy, we show that the only copy of FtsZ in *M. smithii* forms a ring-like structure and co-localizes at mid cell with SepF. A very recent study in *H. volcanii* suggests that FtsZ1 goes first to the division plane followed by SepF that then recruits

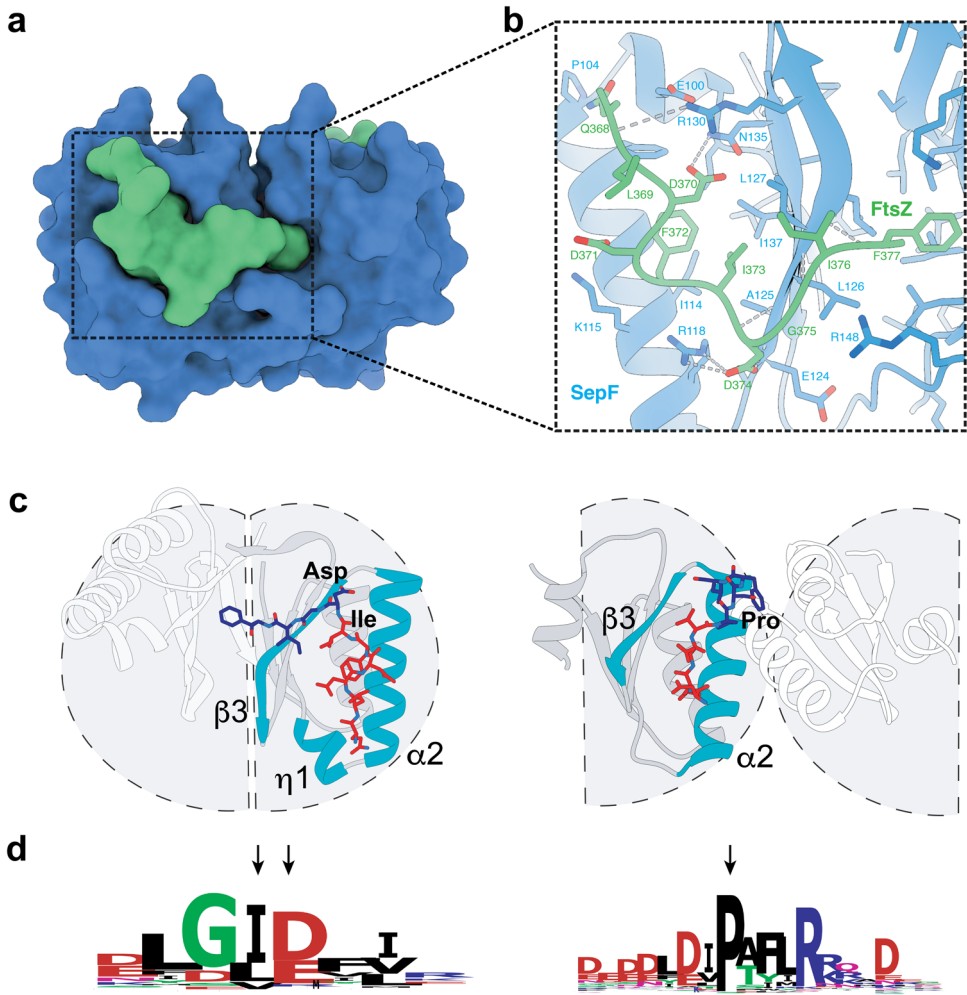

**Fig. 4 The SepF-FtsZ$_{CTD}$ complex.** **a** Surface representation of the crystal structure of FtsZ$_{CTD}$ (green) bound to MsSepF$_{core}$ (blue). **b** Detailed protein-protein interactions within the binding pocket. The side-chains of contact residues ($d < 5$ Å) are shown in stick representation and hydrogen bonds are represented by gray dotted lines. **c** Cartoon representations of the different functional dimers of archaeal MsSepF (left panel) and bacterial CgSepF (right panel) bound to FtsZ$_{CTD}$ (for clarity, only one of the two bound peptides is shown in stick representation). The N-terminal half of FtsZ$_{CTD}$ (in red) binds to a similar groove of the SepF monomer formed between secondary structure elements α2, β3 and η1 (only in MsSepF), shown in turquoise. Highly conserved positions in archaeal and bacterial FtsZ$_{CTD}$ are labeled. **d** Sequence conservation logo of archaeal (left) and bacterial (right) FtsZ$_{CTD}$ sequences. The arrows indicate the highly conserved residues across Archaea or Bacteria that are labeled in panel **c**.

and anchors FtsZ2 to the septum[29]. In *M. smithii* SepF seems to localize to the future division site before FtsZ (Fig. 1d). This is in contrast to what was observed in the ovococcoid bacterium *Streptococcus pneumoniae*, where SepF and FtsZ initially colocalize at the septum, but then FtsZ only relocalizes to the new division sites[30]. However, *S. pneumoniae* also possess the essential alternative Z-ring tether FtsA[30] as well as MapZ, a transmembrane protein that forms a ring-like structure, marks the division site before FtsZ and positions the Z-ring[31]. *M. smithii* SepF may also have a role similar to bacterial MapZ in priming the division site and placing FtsZ. However, unlike MapZ, archaeal SepF is fully cytoplasmic. This would mean that an additional partner may exist in *M. smithii* to help position SepF at mid-cell, but this will need further investigations.

Our results show that SepF-mediated anchoring of FtsZ to the membrane for cytokinesis dates back to billions of years ago, prior to the divergence between Archaea and Bacteria. In agreement with this hypothesis, the crystal structures of the SepF-FtsZ$_{CTD}$ complexes from Bacteria[21] and Archaea (this work) reveal a partially conserved FtsZ-binding pocket in monomeric SepF that might reflect a common ancestral feature. However, the

crystallographic studies as well as the consensus motifs of archaeal and bacterial FtsZ$_{CTD}$ (Fig. 4d) reveal clearly distinctive characteristics, notably at the level of the dimer interface and FtsZ binding. Such major divergence may be linked to fundamental differences in cell envelope and division in the two prokaryotic domains which are worth discussing.

The overall structure of the SepF monomer is very similar in all available archaeal and bacterial crystal structures, and in all cases the functional unit is a dimer. But a notable difference is the wider and deeper binding cleft of the FtsZ binding pocket in MsSepF formed between α2 and β3 through the archaeal-specific conserved η1 insertion. This results in a more important interaction of the FtsZ$_{CTD}$ with the SepF monomer in Archaea than in Bacteria (Fig. 4c) and could point towards an ancestral binding mode where a single monomer of SepF might have been sufficient for FtsZ binding. Interestingly, the functional dimer (as defined by its complex with FtsZ) strikingly differs between MsSepF, where it is formed via a β-β interface, and CgSepF, where it is formed via an α-α interface (Figs. 3b and 4c). Importantly, a bacterial-like α-α interface is not possible in archaeal SepF because a crucial glycine residue (Gly114 in CgSepF) buried at the

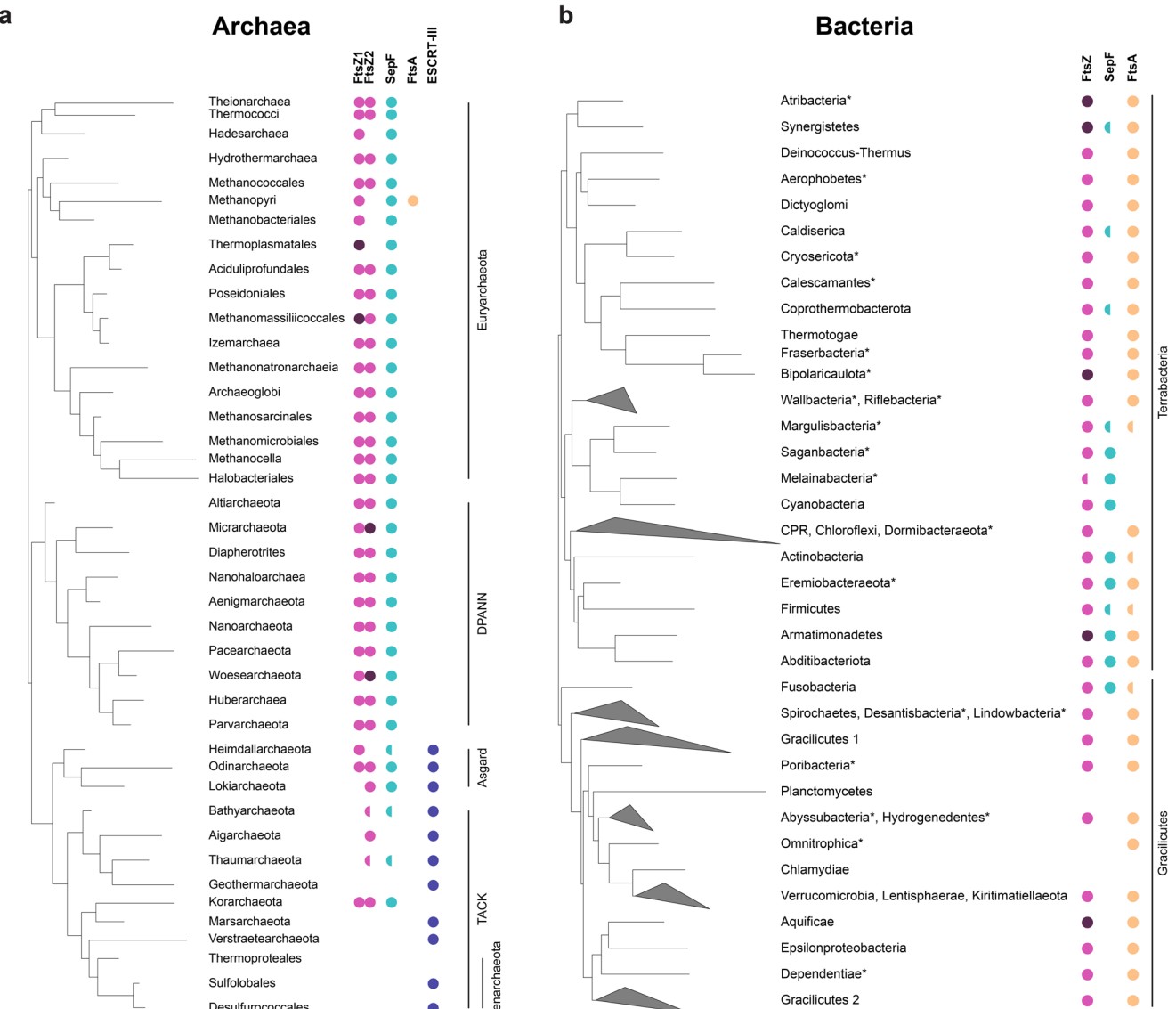

**Fig. 5 SepF is widely present in Archaea and co-occurs with FtsZ. a** Distribution of FtsZ1, FtsZ2, SepF, FtsA and ESCRT-III (CdvB) homologs on a schematic reference phylogeny of the Archaea. FtsZ1 and FtsZ2 homologs are present in most archaeal lineages (magenta), and frequently in two or more copies each (dark magenta). Semicircles indicate that the corresponding protein could not be identified in all the taxa of the corresponding clade and may be either due to true absences or partial genomes. The presence of SepF (turquoise) correlates to that of FtsZ in the majority of taxa. A FtsA homolog was only identified in Methanopyri (orange). Homologs of ESCRT-III (CdvB and homologs) (purple) are only present in the Asgard superphylum and in most representatives of the TACK superphylum, except for Korarchaeota and Thermoproteales. For full data see Supplementary Data 1. **b** Distribution of FtsZ, SepF and FtsA homologs on a schematic reference phylogeny of Bacteria. FtsZ (magenta) is present in most bacterial phyla, with the exception of some phyla within the PVC superphylum (Planctomycetes, Omnitrophica and Chlamydiae), which are known to have specific FtsZ-less cytokinesis. Both SepF and FtsA are also absent in Planctomycetes and Chlamydiae. Two copies of FtsZ can be identified in Atribacteria, Synergistetes, Biopolaricaulota, Armatimonadetes and Aquificae (dark magenta). SepF is present in some phyla (Synergistetes, Caldiserica, Coprothermobacterota, Margulisbacteria, Melainabacteria, Cyanobacteria, Actinobacteria, Eremiobacteraeota, Firmicutes, Armatimonadetes, Abditibacteriota and Fusobacteria), most belonging to the Terrabacteria. In contrast, all Gracilicutes except Fusobacteria, and the remaining Terrabacteria phyla have only FtsA, while Cyanobacteria, Melainabacteria and Saganbacteria have only SepF. Finally, some members of Synergistetes, Caldiserica, Coprothermobacterota, Margulisbacteria, Actinobacteria, Eremiobacteraeota, Firmicutes, Armatimonadetes, Abditibacteriota and Fusobacteria have both FtsA and SepF. Semicircles indicate that the corresponding protein could not be identified in all the analyzed taxa in the lineage displayed. * indicates uncultured Candidate phyla for which many genomes are incomplete. These phyla were not included in phylogenetic reconstructions. For full data see Supplementary Data 2.

center of the bacterial interface is substituted by charged residues with longer side chains (Lys, Glu, Asp) in archaeal sequences (Supplementary Fig. 10), precluding such interaction to occur.

Interestingly, the crystal structure of *B. subtilis* SepF has shown that it can form both bacterial-like α-α and archaeal-like β-β interfaces[22]. The presence of these two self-interacting interfaces has suggested a simple mechanism for *B. subtilis* SepF, where FtsZ

binding occurs through the α-α interface and polymerization through the β-β interface[22], a mechanism which would therefore be absent in Archaea. However, the only other available bacterial SepF structure (*Cg*SepF) revealed that a C-terminal helix α3 interacts with the β-sheet of SepF and would therefore preclude formation of a β-β interface[21]. Remarkably, helix α3 is predicted to be conserved in most bacterial sequences including *B. subtilis*,

but not in Archaea (Supplementary Fig. 7). This observation may suggest that the β-β interface observed in the crystal structure of *B. subtilis* SepF is not involved in bacterial SepF polymerization and might rather be a vestigial feature of the ancestral archaeal-like SepF. Alternatively, by interacting with other cell division partners, the amphipathic helix α3 may be involved in regulating bacterial SepF polymerization through formation of a β-β interface, a possibility that will require further study.

In Bacteria, SepF polymerization has been associated with membrane remodeling[21,22]. Here, we show that *Ms*SepF is also able to tubulate liposome surfaces, although it remains to be determined whether this requires polymerization. In fact, we (and others[29]) did not obtain conclusive experimental evidence that *Ms*SepF polymerizes, and the three available archaeal SepF structures provide no hints about a possible mechanism.

The fundamental divergence between SepF/FtsZ interaction in *M. smithii* and *C. glutamicum* correlates well with the profound differences in the cell envelope structure of Archaea with respect to Bacteria, the former being much more variable and generally lacking a peptidoglycan cell wall[24]. In contrast, the emergence of a rigid cell wall made of peptidoglycan occurred very early in bacterial evolution and likely drove the emergence of the associated complex multi-protein divisome machinery to provide additional mechanical force. Our results suggest that FtsA has also an early origin in Bacteria (Supplementary Fig. 16) and may have provided an alternative tether in addition to SepF, eventually replacing it completely during bacterial diversification. This phenomenon could be linked with the fact that FtsA is a highly dynamic ATP-dependent protein that might therefore have allowed to couple cell division with the energy status of the cell.

The evolution of cytokinesis in Archaea seems to have been less constrained and more dynamic, with (1) the early presence of two FtsZ paralogues and their subsequent independent losses[1,7], (2) the appearance of a multitude of FtsZ-like protein families in some lineages such as Haloarchaea[8,32], and (3) eventually the emergence of the ESCRT-like system and its takeover of cytokinesis in the archaeal lineages closest to the origin of eukaryotes[1,7]. The emergence of a cell wall made of pPG in *M. smithii* and all *Methanobacteriales* and the related *Methanopyrales* is more recent than in Bacteria and is very likely an evolutionary convergence. It will be interesting to investigate if it led to a peculiar SepF/FtsZ interaction by obtaining additional structural data in other wall-less archaeal lineages. Indeed, a single amino acid (Phe377) in FtsZ specifically interacts with the second protomer in the SepF dimer (Fig. 4). This position is only conserved in *M. smithii* related taxa and perhaps provides extra mechanical strength to drive invagination of their thick pPG cell wall.

Finally, our results strongly suggest that a SepF/FtsZ system was already present in the LUCA, unequivocally defining it as a cellular entity. LUCA may not have possessed a complex envelope, and cell division could have been mainly achieved by a minimal system where SepF converted the dynamic energy of FtsZ into mechanical force to achieve membrane constriction[33,34]. Several pieces of evidence suggest that the LUCA only contained a rather basic plasma membrane with the absence of a cell wall[35,36], thus, a membrane-interacting protein bound to a dynamic cytoskeleton element could have represented enough to promote cell division. Therefore, a plausible evolutionary scenario could be that SepF and FtsZ represent the minimal ancestral cell division apparatus in the LUCA, possibly together with a few auxiliary proteins. Given the large conservation of the SepF/FtsZ system in Archaea with respect to Bacteria, combined with its peculiar features, it is tempting to speculate that it may have retained ancestral features dating back to the LUCA.

In conclusion, our results pave the way for future studies to understand cell division in different archaeal models, in particular

gut-associated methanogens with a cell wall made of pPG. FtsZ/SepF interaction represents only part of the story and many questions remain to be answered, such as how the Z-ring is positioned in Archaea, and which other proteins are part of the archaeal divisome. The answers to these questions will allow to reveal fundamental features of contemporary archaeal cell biology, while at the same time dwelling into the most ancient evolutionary past.

## Methods

**Bacterial and archaeal strains and growth conditions**. All bacterial and archaeal strains used in this study are listed in Supplementary table 1. *Escherichia coli* DH5α or Top10 were used for cloning and were grown in Luria-Bertani (LB) broth or agar plates at 37 °C supplemented with 50 μg mL$^{-1}$ kanamycin or 100 μg mL$^{-1}$ ampicillin when required. For protein production, *E. coli* BL21 (DE3) was grown in LB or 2YT broth supplemented with 50 μg mL$^{-1}$ kanamycin or 100 μg mL$^{-1}$ ampicillin at the appropriate temperature for protein expression.

*Methanothermobacter wolfeii* DSM 2970 and *Methanobrevibacter smithii* strain DSM 861 were obtained from the Deutsche Sammlung von Mikroorganismen und Zellkulturen (DSMZ; Braunschweig, Germany). *M. wolfeii* and *M. smithii* were grown approximately two to three weeks in 100 mL serum bottles (clear glass bottle, Supelco) in chemically defined media under strict anaerobic conditions. The gas phase used for both strains was 80 Vol.-% H$_2$ in CO$_2$ at 2.0 bar. *M. wolfeii* was grown at 60 °C and *M. smithii* at 37 °C, both shaking at 180 rpm.

**Methanogen media**. *M. wolfeii* was grown in *Methanothermobacter marburgensis* medium as described in[37]. *M. smithii* was grown in adapted DSMZ 141 Methanogen medium that contained 0.17 g KCl, 2 g MgCl$_2$·6H$_2$O, 0.125 g NH$_4$Cl, 0.053 g CaCl$_2$, 0.055 g KH$_2$PO$_4$, 0.84 g MgSO$_4$, 3 g NaCl, 0.5 g Na-acetate, 1 g yeast extract, 5 mL of trace element solution, 1 mL of FeII(NH$_4$)$_2$(SO$_4$)$_2$·6H$_2$O solution, 50 μL of Na-resazurin solution (0.7 mg mL$^{-1}$), and was filled up with 480 mL of ddH$_2$O. The medium was brought to boil and was boiled for 5 min. After the temperature descended to 50 °C, 5 mL of vitamin solution, 2.5 g NaHCO$_3$, 0.25g L-Cystein-HCl and 0.1 g Na$_2$S·H$_2$O were added. The pH was adjusted to 7 with HCl and the medium was filled up with ddH$_2$O to a final volume of 500 mL. The medium was flushed with CO$_2$ and transferred with a glass syringe into 100 mL serum bottles. 50 mL of the medium were aliquoted into serum bottles and sealed with blue rubber stoppers (pretreated by boiling ten times for 30 min in fresh ddH$_2$O; 20 mm, Bellco) and open-top aluminum caps (9.5 mm opening, Merck Group). The sealed serum bottles containing the medium were autoclaved for 20 min at 120 °C. Composition of trace element solution was: 1.5 g Nitrilotriacetic acid, 3g MgSO$_4$·7H$_2$O, 0.585 g MnCl$_4$·4H$_2$O, 1 g NaCl, 0.1 g FeSO$_4$·7H$_2$O, 0.18 g CoSO$_4$·7H$_2$O, 0.1 g CaCl$_2$·2H$_2$O, 0.18 g ZnSO$_4$·7H$_2$O, 0.006 g CuSO$_4$, 0.02 g KAl (SO$_4$)$_2$·12H$_2$O, 0.01 g H$_3$BO$_3$, 0.01 g Na$_2$MoO$_4$·2H$_2$O, 0.03 g NiCl$_2$·6H$_2$O, 0.3 mg Na$_2$SeO$_3$·5H$_2$O, 0.4 mg Na$_2$WO$_4$·2H$_2$O. First nitrilotriacetic acid was dissolved and pH was adjusted to 6.5 with KOH. Then minerals were added, medium was filled up with ddH$_2$O to a final volume of 1000 mL and pH was adjusted to 7.0 with KOH. Composition of Vitamin solution was: 2 mg Biotin, 2 mg Folic acid, 10 mg Pyridoxine-HCl, 5 mg Thiamine-HCl, 5 mg Riboflavin, 5 mg Nicotinic acid, 5 mg D-Ca-pantothenate, 0.1 mg Vitamin B$_{12}$, 5 mg p-Aminobenzoic acid, 5 mg Lipoic acid and filled up with ddH$_2$O to a final volume of 1000 mL.

**Cloning of *M. wolfeii* PeiW for recombinant protein production in *E. coli***. Genomic DNA from *M. wolfeii* was obtained by extracting with a phenol-chloroform extraction procedure. The gene encoding for PeiW (*psiM100p36*) was amplified by PCR[38] and the sequence of amplified *peiW* was verified by Sanger sequencing (Eurofins Genomics, Ebersberg, Germany). The insert and the Novagen vector pET-15b (Merck Group, Darmstadt, Germany) were digested with NdeI (NEB, Ipswich, MA, USA) and XhoI (NEB, Ipswich, MA, USA) and ligation of digested *peiW* and pET-15b with Quick ligase™ (NEB, Ipswich, MA, USA) generated the plasmid pET-15b_*peiW*, which was transformed into *E. coli* Top10. The successful transformation was confirmed by a colony PCR and by Sanger sequencing (Eurofins Genomics, Ebersberg, Germany). Plasmid DNA was obtained by extracting with a Miniprep Kit (Pure Yield™, Promega, Madison, WI, USA), and pET-15b_*peiW* was transformed into *E. coli* DE3 BL21-AI (Life technologies, Van Allen Way Carlsbad, CA, USA). All plasmids and primers used in this study are listed in Supplementary table 1 and Supplementary table 2 respectively.

**Protein expression and purification of PeiW**. The expression of recombinant PeiW was induced at OD$_{600}$ = 0.6 by the addition of 0.2 % (w/v) L-arabinose and 1 mmol L$^{-1}$ IPTG to LB medium (100 μg mL$^{-1}$ ampicillin). Cell cultures (3 × 250 mL) were harvested after 3 h of incubation by a centrifugation of 15 min at 3170 × g at 4 °C. To verify the expression of recombinant PeiW cell pellets were resuspended to a concentration of 0.01 optical density unit (ODU) μL$^{-1}$ in 5x Laemmli buffer and 1 μL DTT (1 mol L$^{-1}$) and lysed 5 min at 95 °C and 1050 rpm. Crude extracts were centrifuged 30 min at 16,100 × g at 4 °C and supernatant was loaded on two 12.5 % (w/v) polyacrylamide gels. The gel was stained with Commassie Brilliant Blue (50% (v/v)

ddH$_2$O, 40 % (v/v) ethanol (ethanol absolute), 10 % (v/v) 100 Vol.-% acetic acid, 0.1 % (w/v) R-250 Brilliant Blue).

For protein purification frozen cell pellets were resuspended in Ni-NTA lysis buffer (50 mmol L$^{-1}$ NaH$_2$PO$_4$, 300 mmol L$^{-1}$ NaCl, 10 mmol L$^{-1}$ imidazole, at pH 8) at 4 °C and lysed by sonication. The lysate was centrifuged for 20 min at 16.9x g at 4 °C. The cleared lysate was incubated for 2 h at 4 °C with Ni-NTA agarose resin (Invitrogen) under slight agitation. The lysate with the beads was passed through a gravity flow chromatography column (Econo-Pac chromatography columns, Biorad). The column was washed with four column volumes of washing buffer (50 mmol L$^{-1}$ NaH$_2$PO$_4$, 300 mmol L$^{-1}$ NaCl, 20 mmol L$^{-1}$ imidazole, at pH 8). His-tagged proteins were eluted with 3 × 1 mL elution buffer (50 mmol L$^{-1}$ NaH$_2$PO$_4$, 300 mmol L$^{-1}$ NaCl, 250 mmol L$^{-1}$ imidazole, pH = 8) and fractions were collected separately. The fractions containing the protein of interest were loaded on a PD-midi Trap G25 column (GE Health Care) and eluted with 1.5 mL of 50 mmol L$^{-1}$ Hepes buffer (at pH 7) for buffer exchange.

**Cloning of *M. smithii* SepF and FtsZ for recombinant protein production in *E. coli*.** Genomic DNA from *M. smithii* was obtained by resuspending a pellet of densely grown culture in ddH$_2$O and boiling it for 5 min at 99 °C. Subsequently, 1 μL of archaeal suspension was used as template in each 50 μL PCR reaction to amplify the genes encoding for *M. smithii ftsz* (MSM_RS03130), *sepF* full length (SepF, MSM_RS02010) as well as only the core region of *sepF* (SepF$_{core}$) comprising amino acids from 54 to 149. The products of the right size were cloned in to the pT7 vector containing an N-terminal 6xHis-SUMO tag by Gibson assembly. The constructs were transformed into chemically competent DH5 α *E. coli* cells. The successful transformation was confirmed by a colony PCR and by Sanger sequencing (Eurofins Genomics, France).

**Protein expression and purification.** Protein expression and purification was carried out as previously describes in[21]. In brief, N-terminal 6xHis-SUMO-tagged SepF and SepF$_{core}$ from *M. smithii* were expressed in *E. coli* BL21 (DE3). After 4 h at 37 °C cells were grown for 24 h at 20 °C in 2YT complemented auto-induction medium[39] containing 50 μg mL$^{-1}$ kanamycin. Cell pellets were resuspended in 50 mL lysis buffer (50 mmol L$^{-1}$ Hepes pH8, 300 mmol L$^{-1}$ NaCl, 5% (v/v) glycerol, 1 mmol L$^{-1}$ MgCl$_2$, benzonase, lysozyme, 0.25 mmol L$^{-1}$ TCEP, EDTA-free protease inhibitor cocktails (ROCHE)) at 4 °C and lysed by sonication. The lysate was centrifuged for 30 min at 30,000x g at 4 °C and loaded onto a Ni-NTA affinity chromatography column (HisTrap FF crude, GE Healthcare). His-tagged proteins were eluted with a linear gradient of buffer B (50 mmol L$^{-1}$ Hepes at pH 8, 300 mmol L$^{-1}$ NaCl, 5 % (v/v) glycerol, 1 mol L$^{-1}$ imidazole). The eluted protein of interest was dialyzed at 4 °C overnight in SEC buffer (50 mmol L$^{-1}$ Hepes at pH 8, 150 mmol L$^{-1}$ NaCl, 5 % (v/v) glycerol) in the presence of the SUMO protease (ratio used, 1:100). The cleaved protein was concentrated and loaded onto a Superdex 75 16/60 size exclusion (SEC) column (GE Healthcare) pre-equilibrated at 4 °C in 50 mmol L$^{-1}$ Hepes at pH 8, 150 mmol L$^{-1}$ NaCl, 5 % (v/v) glycerol. The purified protein was concentrated, aliquoted, flash frozen in liquid nitrogen and stored at −80 °C.

N-terminal 6xHis-SUMO-tagged *M. smithii* FtsZ was produced and purified as described above, except induction was performed at 37 °C, KCl was used instead of NaCl in all the buffers and a TALON FF crude column (GE Healthcare) was used for affinity chromatography.

**Thermal shift assay.** Tests were performed using 96 well plate and 20 μL of reaction volume per well. A total of 3 μg of *Ms*SepF$_{core}$ was used per assay in a buffer containing 150 mmol L$^{-1}$ NaCl, 25 mmol L$^{-1}$ Hepes at pH 8 and 5% (v/v) glycerol. When used, FtsZ$_{CTD}$ peptide (NEDQLDDFIDGIF purchased from Genosphere) was added to a final concentration of 1 mmol L$^{-1}$. Next, 0.6 μL of 50X Sypro Orange solution (Invitrogen) was added to each well and samples were heated from 25 to 95 °C in 1 °C steps of 1 min each in a CFX96 Touch™ Real-Time PCR Detection System (BioRad). Excitation/emission filters of 492 and 516 nm were used to monitor the fluorescence increase resulting from binding of the Sypro Orange to exposed hydrophobic regions of the unfolding *Ms*SepF$_{core}$. The midpoint of the protein unfolding transition was defined as the melting temperature (Tm).

**Surface plasmon resonance (SPR) assay.** SPR experiments were performed on a Biacore T200 instrument (Cytiva) equilibrated at 25 °C in Biacore Buffer buffer (150 mmol L$^{-1}$ NaCl, 25 mmol L$^{-1}$ Hepes pH8). For the *Ms*SepF$_{core}$ covalent immobilization, two flowcells of a CM5 sensorchip (Cytiva) were covalently functionalized using NHS/EDC amine coupling chemistry. The carboxymethylated dextran surface was first activated by flowing a mixture of NHS (50 mmol L$^{-1}$) and EDC (200 mmol L$^{-1}$) for 10 min at 5 μL min$^{-1}$. *Ms*SepF$_{core}$ (diluted in 10 mmol L$^{-1}$ acetate pH4) was then injected on one flowcell for 20 min at 6 μg mL$^{-1}$ and on another for 30 min at 30 μg mL$^{-1}$. Both surfaces were then de-activated by injecting ethanolamine 1 mol L$^{-1}$ for 10 min. The final densities of *Ms*SepF$_{core}$ were respectively 3200 RU and 5000 RU (1RU~1 pg/mm$^2$). For Real-time monitoring of the interaction between the FtsZ$_{CTD}$ peptide and *Ms*SepF$_{core}$, 9 concentrations of the FtsZ$_{CTD}$ peptide (ranging from 3.9 to 1000 nmol L$^{-1}$) were flowed in triplicate for 30 s at 20 μL min$^{-1}$ over the two covalently-immobilized

*Ms*SepF$_{core}$ surfaces. The dissociation of the peptide/*Ms*SepF$_{core}$ complexes was then monitored for 240 s by flowing buffer over the two flowcells. The dissociation equilibrium constant of the interaction (Kd) was determined by analizing the concentration-dependence of the steady-state SPR signals (Req) using the following equation: Req = Rmax × C/Kd + C (where C is the FtsZ$_{CTD}$ peptide concentration and Rmax the SPR response at saturating peptide concentration).

**Crystallization.** Crystallization screens were set up for *Ms*SepF$_{core}$ apo form and in complex with FtsZ$_{CTD}$ (molar ratio 1:5) using the sitting-drop vapor diffusion method at 18 °C in a Mosquito nanoliter-dispensing crystallization robot (TTP Labtech, Melbourn, UK) as detailed in[40]. Optimal crystals of *Ms*SepF$_{core}$ (14 mg mL$^{-1}$) appeared after 5 days in a buffer containing 0.1 mol L$^{-1}$ TRIS at pH 8.5 and 30 % (w/v) PEG 10 K. The complex *Ms*SepF$_{core}$-FtsZ$_{CTD}$ crystallized in a buffer containing 0.2 mol L$^{-1}$ (NH$_4$)$_2$SO$_4$ and 30 % (w/v) PEG 8 K after 1 week using *Ms*SepF$_{core}$ at 14 mg mL$^{-1}$ and FtsZ$_{CTD}$ at 6 mg mL$^{-1}$. Crystals were cryo-protected in the crystallization buffer containing 33% (v/v) glycerol.

**Data collection, structure determination and refinement.** X-ray diffraction data were collected at 100 K using beamlines Proxima-1 and Proxima-2 at the Soleil synchrotron (GIF-sur-YVETTE, France). Both data sets were processed using XDS[41] and AIMLESS from the CCP4 suite[42]. The crystal structures were solved by molecular replacement using Phaser[43] and a single monomer of the PDB 3ZIE as a search model. All structures were refined through several cycles of manual building with COOT[44] and reciprocal space refinement with BUSTER[45]. Both final models exhibit a good stereochemistry, with no Ramachandran outliers. The Table 1 shows the crystallographic statistics. Structural figures were generated using Chimera[46]. Atomic coordinates and structure factors can be found in the protein data bank under the accession codes 7AL1 and 7AL2.

**Small unilamellar vesicles (SUVs) preparation.** SUVs were prepared as previously describes in[21]. In brief, reverse phase evaporation was used. A 10 mmol L$^{-1}$ lipids chloroform solution was prepared and chloroform was removed by evaporation under vacuum conditions. The dried phospholipid film was resuspended in a mixture of diethyl ether and 25 mmol L$^{-1}$ Hepes buffer at pH 7.4 and remaining diethyl ether was eliminated by reverse phase evaporation. Finally, SUVs were obtained by sonication during 30 min at 4 °C.

**Lipid peptide interaction (tryptophan fluorescence emission titration).** To estimate the partition coefficient (K$_x$) between SUVs and SepF a similar protocol as described in[21] was used. In brief, we used the synthetic peptide WMGFTDALKRSLGF (purchased from Genosphere), which contains the SepF$_M$ sequence of *M. smithii* with an extra W residue at the N-terminal. K$_x$ is defined as the ratio of peptide concentration in the lipid and in the buffer phases. K$_x$ can be expressed by the following equation:

$$K_X = \frac{P_L/(P_L + [L])}{P_W/(P_W + [W])}$$

in which P$_W$ represent the concentration of soluble peptide (in aqueous phase) and P$_L$ the peptide concentration bound to lipid membranes (lipidic phase). [L] refers to the lipid concentration and [W] refers to the water concentration. K$_x$ is directly related to the apparent dissociation constant as K$_x$ * Kd = [W] with Kd * P$_L$ = P$_W$ × [L]. The KaleidaGraph software was used to fit the K$_x$ to the experimental data.

We used a FP-8200 (Jasco, Tokyo, Japan) spectrophotometer equipped with a thermostatic Peltier ETC-272T at 25 °C. Experiments were performed in a high precision cell cuvette made of Quartz with a light Path of 10 × 4 mm (Hellma, France). A bandwidth of 5 nm was used for emission and excitation. 1 μmol L$^{-1}$ of peptide was used in Titration buffer (100 mmol L$^{-1}$ KCl and 25 mmol L$^{-1}$ Hepes at pH 7.4). We measured the florescence emission between 300 and 400 nm at a scan rate of 125 nm min$^{-1}$ with an excitation wavelength of 280 nm. The obtained spectra were corrected by blank subtraction (SUV light scattering in titration buffer). Next, the maximum wavelength value (λ$_{max}$) was calculated to measure the partition coefficient (K$_x$).

**SepF polymerization assay.** Purified *Ms*SepF and *Ms*SepF$_{core}$ were precleared at 25,000 *g* for 15 min at 4 °C. To follow the polymerization of *Ms*SepF in the presence of lipids, we used *Ms*SepF or *Ms*SepF$_{core}$ plus SUVs at a final concentration of 50 μmol L$^{-1}$ each in SepF polymerization buffer containing 100 mmol L$^{-1}$ KCl, 10 mmol L$^{-1}$ MgCl$_2$ and 25 mmol L$^{-1}$ Pipes at pH 6.9. The mixture was placed into a quartz cuvette with a light path of 10 mm. Data acquisition started immediately using an UV–Visible Spectrophotometer (Thermo scientific Evolution 220) during 600 s at 25 °C using 400 nm for excitation and emission and spectra with slits widths of 1 nm. Measurements were taken every 15 s during 600 s and keeping a constant temperature of 25 °C.

**Electron microscopy.** For negative stain sample preparations, incubations were performed at room temperature. SUVs (100 μmol L$^{-1}$) alone or in the presence of SepF (50 μmol L$^{-1}$) with or without FtsZ$_{CTD}$ (100 μmol L$^{-1}$) were incubated in a buffer containing 100 mmol L$^{-1}$ KCl, 10 mmol L$^{-1}$ MgCl$_2$ and 25 mmol L$^{-1}$ Pipes

at pH 6.9 for 10 min. In order to visualize FtsZ filaments, FtsZ (30 µmol L$^{-1}$) was incubated with or without SepF$_{core}$ (20 µmol L$^{-1}$) in polymerization buffer (2 mol L$^{-1}$ KCl, 50 mmol L$^{-1}$ Hepes at pH 7.4 and 10 mmol L$^{-1}$ MgCl$_2$) supplemented with freshly prepared 0.6 mol L$^{-1}$ Trimethylamine N-oxide (TMAO) and 3 mmol L$^{-1}$ GTP. Reactions were incubated for 10 min before applied onto the grid.

For all samples, 400 mesh carbon coated grids (Electron Microscopy Sciences; CF 400-Cu) were glow-discharged on an ELMO system for 30 sec at 2 mA. 5 µL of sample was applied onto the grid and incubated for 30 s, the sample was blotted, washed in three drops of buffer (100 mmol L$^{-1}$ KCl, 10 mmol L$^{-1}$ MgCl$_2$ and 25 mmol L$^{-1}$ Pipes at pH 6.9) and then stained with 2% (w/v) uranyl acetate. Images were recorded on a Gatan UltraScan4000 CCD camera (Gatan) on a Tecnai T12 BioTWINLaB6 electron microscope operating at a voltage of 120 kV.

**Statistical analysis of FtsZ filament diameter.** Electron microscopy images of negatively stained FtsZ filaments (FtsZ + GTP) and FtsZ bundles (FtsZ + GTP + SepF$_{core}$) were analyzed using the public domain program Fiji (ImageJ) Version 2.0.0-rc-68/1.52i. The diameter of each 130 FtsZ filaments and FtsZ bundles was measured in several images each and the data were extracted into Microsoft Excel 2016. Boxplots were created by using Microsoft Excel 2016 with the Real Statistics Resource Pack for Excel 2016 (http://www.real-statistics.com). Each data set for diameter of FtsZ filaments and FtsZ bundles was tested for normal distribution using the Kolmogornov-Smirnov test (KS-test). The KS-statistic was 0.9798 (p = 5.5E−0111) for FtsZ filaments and 1.0000 (p = 1.3E−0115) for FtsZ bundles. For a level of significance of 5%, critical values of 0.1178 and 0.1178 were obtained, respectively. Hence, the data are normal distributed in both cases, because the null hypothesis (H$_0$) was not rejected, as the critical values are >α. A 2-sample t-test was conducted to test whether the means of the diameter of FtsZ filaments and FtsZ bundles are significantly different. The result for the t-statistic was found to be −23.8032. The 2-sample *t*-test resulted in a critical t-value of t(alpha) of 1.9692 for an alpha of 0.025. α/2 is to be used for a two sample two tailed test. Thus, a significant difference of the diameter was found at a 5% level of significance, because the H$_0$ was rejected, as the modulus of the critical value was > α/2. The statistical analysis was performed using DataLab version 4.0 (Epina GmbH, Pressbaum, Austria).

**Western blots and SepF and FtsZ quantification.** Purified *Ms*SepF$_{core}$ protein was used to raise antibodies in guinea pig (Covalab). Two peptides (CEN-AENGLEKLKSAADT and CGESDSGDRALESVHE) in equimolar amounts were used to raise antibodies in rabbit against *M. smithii* FtsZ (Genosphere Biotech). To prepare cell extracts, archaeal cell pellets were resuspended in PBS buffer (137 mmol L$^{-1}$ NaCl, 10 mmol L$^{-1}$ Na$_2$HPO$_4$, 1.8 mmol L$^{-1}$ KH$_2$PO$_4$, 2.7 mmol L$^{-1}$ KCl, at pH 7.4) together with LDS sample buffer (Invitrogen) and 100 mmol L$^{-1}$ DTT, and disrupted at RT with 0.1 mm glass beads and using the MP FastPrep-24™ 5 G homogenizer. In order to test the expression of the proteins and quantify the amount of SepF and FtsZ in cells, crude extracts and purified recombinant SepF and FtsZ in serial dilution were loaded on a 4–12 % (w/v) Bis-Tris PAGE gel (Invitrogen). The amounts of crude extract were 120 µg for FtsZ and 290 µg for SepF. Purified FtsZ protein was serially diluted from 100 ng to 6.26 ng, and purified SepF protein from 6.25 ng to 0.1 ng. After separation of proteins on SDS-PAGE gel, they were electro-transferred onto a 0.2 µm Nitrocellulose membrane. Membranes were blocked with 5 % (w/v) skimmed milk for 45 min at room temperature (RT). Membranes were incubated with either an anti-*Ms*SepF antibody or an anti-*Ms*FtsZ antibody, both at 1:500 dilution for 1 h at RT. After washing in TBS-Tween buffer (10 mmol L$^{-1}$ Tris-HCl at pH 8; 150 mmol L$^{-1}$ NaCl; Tween 20 0.05 % (v/v), the membranes were incubated with an anti-rabbit horseradish peroxidase-linked antibody for FtsZ and an anti-guinea pig horseradish peroxidase-linked antibody for SepF (Invitrogen), both at 1:5000 dilution for 45 min. The membranes were washed and revealed with HRP substrate (Immobilon Forte, Millipore) and imaged using the ChemiDoc MP Imaging System (BIORAD). Quantification of the bands were performed using the software Image Lab (Biorad) and the band volume was plotted against the ng loaded in order to obtain a strand curve. All uncropped blots are shown in Supplementary Fig. 1.

**Immunostaining.** Cells of *M. smithii* grown to early exponential phase were harvested by 5 min of centrifugation at 3.5 × g (all centrifugation steps were performed at 3.5 × g). Pellets were washed in PBS buffer, pelleted, fix with 100% ice cold methanol and stored at −20 °C. Cells were rehydrated and washed in 50 mmol L$^{-1}$ HEPES buffer at pH 7 for 10 min at RT, followed by permeabilization of the archaeal pPG by incubation for 10 min with 3.5 µg of purified 6xHis-PeiW in 50 mmol L$^{-1}$ HEPES buffer containing 1 mmol L$^{-1}$ DTT at 71 °C. After the incubation, cells were allowed to cool on ice for 2 min. Cells were pelleted and washed one time in 50 mmol L$^{-1}$ HEPES buffer and two times in PBS-T (PBS buffer with Tween 20 at 0.1 % (v/v)). Blocking was carried out for 1 h in PBS-T containing 2 % (w/v) bovine serum albumin (blocking solution) at RT. Cells were incubated with a 1:200 dilution of rabbit polyclonal anti-*Ms*FtsZ peptide antibody and 1:200 dilution of guinea pig polyclonal anti-*Ms*SepF antibody overnight at 4 °C in blocking solution. Upon incubation with primary antibody samples were pelleted and washed three times in PBS-T, and incubated with a 1:500 dilution of secondary Alexa555-conjugated anti-rabbit for FtsZ and Alexa488-conjugated anti-

guinea pig for SepF (Invitrogen) in blocking solution for 1 h at RT. Unbound secondary antibody was removed by three washing steps in PBS-T. Finally, cells were resuspended in few µL of PBS and slides were prepared either for super resolution microscopy or epifluorescence microscopy (see below).

**Three-dimensional structured illumination microscopy (3D SIM) imaging and analysis.** Archaeal cell suspensions were applied on high precision coverslips (No. 1.5H, Sigma-Aldrich) coated with 0.01 % (w/v) of Poly-L-Lysin. After letting the cells attach onto the surface of the coverslip for 10 min, residual liquid was removed, 8 µL of antifade mounting medium (Vectashield) were applied and the coverslip was sealed to a slide. SIM was performed on a Zeiss LSM 780 Elyra PS1 microscope (Carl Zeiss, Germany) using C Plan-Apochromat 63×/1.4 oil objective with a 1.518 refractive index oil (Carl Zeiss, Germany). The samples were excited with laser at 488 nm and 561 nm and the emission was detected through emission filter BP 495–575 + LP 750 and BP 570–650 + LP 750, respectively. The fluorescence signal was detected on an EMCCD Andor Ixon 887 1 K camera. Raw images are composed of fifteen images per plane per channel (five phases, three angles), and acquired with a Z-distance of 0.10 µm. Acquisition parameters were adapted from one image to another to optimize the signal to noise ratio. SIM images were processed with ZEN software (Carl Zeiss, Germany) and then corrected for chromatic aberration using 100 nm TetraSpeck microspheres (ThermoFisher Scientific) embedded in the same mounting media as the sample. The Fiji plugin SIMcheck was used to analyze the quality of the acquisition and the processing in order exclude imaging artifacts[47]. The analysis was performed on the whole field of view imaged and results of the SIMcheck are shown in Supplementary table 3. For further image analysis of SIM image z stacks we used Fiji (ImageJ) Version 2.0.0-rc-68/1.52i. Namely, we assigned a color to the fluorescent channel, stacks were fused to a single image (z projection, maximum intensity), stacks were rotated 90° (resliced) prior z projection for the side view, and movies were created via 3D projection. Regions of interest were cut out and, for uniformity, placed on a black squared background. Figures were compiled using Adobe Illustrator 2020 (Adobe Systems Inc. USA).

**Morphometric and fluorescence measurements.** 2 µL of immunolabelled *M. smithii* cells solution together with 1 µL of Vectashield (Vector Labs) were applied to an 1 % (w/v) agarose covered microscopy slide and imaged using a Zeiss Axioplan 2 microscope equipped with an Axiocam 503 mono camera (Carl Zeiss, Germany). Epifluorescence images were acquired using the ZEN lite software (Carl Zeiss, Germany) and processed using Fiji (ImageJ) Version 2.0.0-rc-68/1.52i in combination with plugin MicrobeJ Version 5.13 l. The cell outlines were traced and cell length, width, fluorescence intensity along the cell length and presence of fluorescent maxima were measured automatically. Automatic cell recognition was manually double-checked. For the mean fluorescence intensity plots cells were automatically grouped into four classes according to the detected FtsZ (0–3 fluorescent maxima detected; the data of 0 and 1 fluorescent maxima detected were pooled) and the corresponding FtsZ and SepF mean fluorescence intensity of each group was plotted against the normalized cell length. For the plots showing the relative position of detected maxima, cells were automatically grouped into four classes for SepF (1–4 fluorescent maxima detected) and three for FtsZ (1–3 fluorescent maxima detected). Both, fluorescence intensity and relative position of detected maxima plots were created in MicrobeJ, representative cells were chosen from bigger fields of few and their brightness and contrast were adapted in Fiji. Figures were compiled using and Illustrator 2020 (Adobe Systems Inc. USA).

**Sequence analysis.** The C-terminal domain was defined as the conserved regions flanking the 'GID' motif in Archaea and the 'PAFLR' motif in Bacteria. Noisy columns in the alignments were removed for clarity. Once defined, the C-terminal was realigned with MAFFT using the L-INS-i algorithm[48] and columns with >70% gaps removed with trimAL[49]. WebLogos[50] correspond to alignments of 181 bacterial FtsZ, 117 FtsZ1 and 111 FtsZ2 from Archaea. The low number of well annotated FtsZ2 sequences made us further reduce redundancy of the bacterial dataset, for comparative purposes.

For structure-based SepF alignment, representative sequences from Bacteria and Archaea were manually chosen and aligned using T-Coffee[51]. Graphical representation was made using ENDscript server[52] and manually edited. For secondary structure prediction, sequences were aligned using MAFFT with the L-INS-i algorithm[48] and secondary structures were detected using Ali2D[53]. Figures were compiled using Illustrator 2020 (Adobe Systems Inc. USA).

**Distribution, phylogeny, and synteny analysis.** A customized local genome database was compiled for 362 Bacteria and 150 Archaea representative of all major phyla in NCBI (as of January 2020). The presence of FtsZ, FtsA, SepF and ESCRT-III (CdvB) homologs was assessed by performing HMM-based homology searches using custom made HMM models and the HMMER package[54,55] (default parameters). Separate searches were carried out with archaeal and bacterial queries and absences were checked with TBLASTN[56] or individual HMMER searches against all available taxa in NCBI corresponding to those phyla. Results were mapped onto the bacterial and archaeal reference phylogenies using iTOL[57]. The schematic reference phylogeny of Bacteria and Archaea is based on phylogenies constructed

by using a concatenation of RNA polymerase subunits B, B′ and IF-2 in the case of Bacteria and of phylosift markers (41 markers as in[1]) for the Archaea.

For phylogenetic analysis, FtsZ, FtsA and SepF homologs were aligned using MAFFT (L-INS-i algorithm)[58]. Columns with excess gaps (>80% for FtsA/SepF, >70% for FtsZ) were removed with trimAl[49]. Maximum likelihood trees were generated using IQ-TREE v1.6.7.2[59], using the SH-aLRT branch test ('-alrt 1000' option) and the ultrafast bootstrap approximation[60] (option '–bb 1000') for branch supports. Exchangeability matrices and overall evolutionary models for each alignment were determined using ModelFinder[61], with heterogeneity rates modeled with the gamma distribution (four rate categories, G4), or free-rate models (+R5 to +R9, depending on the protein). For the analysis of bacteria a subset of 190 taxa was used.

To carry out synteny analysis, for each archaeal genome, ten genes upstream and downstream *ftsZ1* and *ftsZ2* and *sepF* were extracted and the corresponding proteins were subjected to all-vs-all pairwise comparisons using BLASTP v2.6.0[56] with default parameters. From the output of the BLASTP search, protein families were assembled with SILIX v1.2.951[62] with default parameters. HMM profiles were created for the families containing members of three or more archaeal lineages using the HMMER package[55]. The families were annotated manually by using BLASTP v2.6.0[56] and the Conserved Domain Database from NCBI[63]. MacSyFinder[64] was then used to identify, in each archaeal taxon, clusters containing at least three genes with a separation no greater than five other genes.

**Reporting summary**. Further information on research design is available in the Nature Research Reporting Summary linked to this article.

## Data availability

The crystallographic data is available from the Protein Data Bank (www.rcsb.org), under the accession numbers 7AL1 and 7AL2 (PDB code, https://doi.org/10.2210/pdb7AL1/pdb; https://doi.org/10.2210/pdb7AL2/pdb). The data used for phylogenetic analysis and alignments is found here: https://data.mendeley.com/datasets/pz8893jzgk/draft?a=5a5da375-729f-44b3-83a2-ca2adf6675d6. All other data are available from the corresponding authors upon reasonable request. Source data are provided with this paper.

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

## Acknowledgements

This work was partially supported by grants from the Institut Pasteur (Paris), the CNRS (France) and the Agence Nationale de la Recherche (PhoCellDiv, ANR-18-CE11-0017-01 and ArchEvol, ANR-16-CE02-0005-01). N.P. is supported by a Pasteur-Roux Postdoctoral Fellowship from the Institut Pasteur (Paris). A.S. and D.M. are part of the Pasteur - Paris University (PPU) International PhD Program, funded by the European Union's Horizon 2020 research and innovation program under the Marie Sklodowska-Curie grant agreement No 665807. M.G. thanks support from Programa de Desarrollo de las Ciencias Básicas (PEDECIBA) and Agencia Nacional de Investigación e Innovación (ANII), Uruguay. We thank A. Chenal for help with the lipid interaction studies. We gratefully acknowledge the core facilities at the Institut Pasteur C2RT, A. Haouz, P. Weber, C. Pissis (PFC). We thank the staff of the synchrotron SOLEIL for assistance and support in using beamlines PX1 and PX2. We thank A. Salle for help with the 3D SIM. We gratefully acknowledge the UTechS Photonic BioImaging (Imagopole), C2RT, Institut Pasteur (Paris, France) as well as the France–BioImaging infrastructure network supported by the French National Research Agency (ANR-10-INSB-04; Investments for the Future), and the Région Ile-de-France (program Domaine d'Intérêt Majeur-Malinf) for the use of the Zeiss LSM 780 Elyra PS1 microscope. We acknowledge technical assistance by Barbara Reischl.

## Author contributions

N.P. performed immunolabelling, epifluorescence microscope and 3D SIM imaging and image analysis. N.P. and A.S. conducted the protein biochemistry and purification for structural and biophysical studies. N.P. and A.S. carried out the biochemical and biophysical studies of protein-protein interactions. A.S. carried out binding studies of lipid membrane-protein interactions. D.M. and M.G. performed sequence and phylogenetic analyses. H.P. and SK.-M.R.R. carried out PeiW purification and provided material. A.S. and P.M.A. carried out the crystallogenesis and crystallographic studies. A.M.W. and A.S.R. performed the negative stain EM studies. P.E. performed the SPR assay. N.P. and S.K.-M.R.R. performed statistical analysis. N.P., A.S and D.M. made the figures. A.M.W., P.M.A and S.G supervised the work. N.P., A.S., A.M.W., P.M.A. and S.G. wrote the manuscript. All authors edited the final version of the manuscript.

## Competing interests

The authors declare no competing interests.
