## [Peer Review File · Nature Communications]

Editorial Note: This manuscript has been previously reviewed at another journal that is not operating a transparent peer review scheme. This document only contains reviewer comments and rebuttal letters for versions considered at *Nature Communications*. Mentions of the other journal have been redacted.

REVIEWERS' COMMENTS

Reviewer #1 (Remarks to the Author):

I am satisfied with the core of the answers and corrections. I will just request the authors to include the controls cited in this response to my and reviewer 2 inquiries. I don't think it's necessary to go through another review round, but I do think it's important to show controls not only to educate our future scientists but also to secure the work of any possible silly questioning. I understand how many review rounds (and transfer between journals) can generate stress to authors, but I judge it to be important to show how careful and dedicated the authors were.

I personally disagree with some of the discussion points in the text (mainly regarding the overlapping anchoring function of FtsA and SepF) but I admit this is a muddy field so I won't let my own biases stop the broader discussion that should happen among the thousands of bacterial and archaeal cell biologists. I consider this discussion yet another contribution from the paper to the community, together with the novelty nature of the data: 1) a new dynamic behavior of a SepF homolog in archaea, showing very distinct localization hierarchy compared to the bacterial counterpart, 2) the first cytokinetic ring anchor in Archaea, and 3) the establishment of a new relevant non-model archaeal cell biology organism. The very depth of the discussion that emerged from the reviews speaks for the richness of the data offered by the authors.

I planned to only address the situation by a direct message to the editor, but now I feel authors should read this, and maybe if they choose to make the reviews public, this could be a useful insight to other young scientists. It is impossible to not feel empathetic to the authors (especially the ECRs powering the experiments) when reviewer 2 seems to think that "SepF in the archaeon *Methanobrevibacter smithii* behaves largely the same as the bacterial equivalent" and "A better mechanistic insights into this peculiar SepF localization mechanism seems reasonable for a [Redacted] paper". I personally don't care if the paper gets published at [Redacted] or Nat Comm. That is not in my mind while serving as a scientific expert to help the paper get to the best shape possible in the least amount of time. Regardless, there are so many (subtle but not less important) mechanistic insights that set bacterial and archaeal biology apart here. The cell structure, having S-layer or pseudomurein, already poses a completely different puzzle both in the cell architecture as in the evolutionary aspect. In my opinion, this is a paper that shows that studying divergent, close to impossible to domesticate organisms will pay off in future generations.

Reviewer #2 (Remarks to the Author):

I have reviewed the Pende et al paper for [Redacted] and my main conclusion was that the mechanistic novelties, in my opinion, were insufficient for this high-impact journal since the results more or less confirm what has been shown in earlier work with bacteria and archaea, and although there are clear differences (e.g. no polymerisation of SepF, at least in vitro), I find it difficult to appreciate these differences as being such major breakthroughs. The authors did a lot of work and established novel cell biology protocols for this archaea, but is this the same as an important mechanistic novel finding? my gut feeling says no. Now the problem is that I consider Nature Communications also a high-impact journal. Therefore, I remain of the opinion that this work does not contribute a level of novelty that really changes our view of how SepF works.

Reviewer #3 (Remarks to the Author):

This manuscript has greatly benefited from the revisions made by the authors and the careful way that most of the comments of the reviewers were addressed. As I have reviewed this manuscript previously and am satisfied with the overall modifications made I have no further remarks and hope to see this paper published shortly. I would like to congratulate the authors on a fine piece of work.

Reviewer #5 (Remarks to the Author):

The revised manuscript by Pende, Sogues et al has been largely improved by inclusion of the different comments and suggestions by the reviewers. From the structural point of view, description of the interactions between components of this unique SepF:FtsZ complex is stronger than before and provide all details required to understand mechanistic differences between bacteria and archaea cases. I also appreciate the effort in including new figures. All my previous concerns have been largely answered in this revised version and I want just felicitate the authors for the important piece of work on this very relevant topic.

REVIEWERS' COMMENTS

Reviewer #1 (Remarks to the Author):

I am satisfied with the core of the answers and corrections. I will just request the authors to include the controls cited in this response to my and reviewer 2 inquiries. I don't think it's necessary to go through another review round, but I do think it's important to show controls not only to educate our future scientists but also to secure the work of any possible silly questioning. I understand how many review rounds (and transfer between journals) can generate stress to authors, but I judge it to be important to show how careful and dedicated the authors were.

AU: we have now included this control as supplementary figure 4.

I personally disagree with some of the discussion points in the text (mainly regarding the overlapping anchoring function of FtsA and SepF) but I admit this is a muddy field so I won't let my own biases stop the broader discussion that should happen among the thousands of bacterial and archaeal cell biologists. I consider this discussion yet another contribution from the paper to the community, together with the novelty nature of the data: 1) a new dynamic behavior of a SepF homolog in archaea, showing very distinct localization hierarchy compared to the bacterial counterpart, 2) the first cytokinetic ring anchor in Archaea, and 3) the establishment of a new relevant non-model archaeal cell biology organism. The very depth of the discussion that emerged from the reviews speaks for the richness of the data offered by the authors.

I planned to only address the situation by a direct message to the editor, but now I feel authors should read this, and maybe if they choose to make the reviews public, this could be a useful insight to other young scientists. It is impossible to not feel empathetic to the authors (especially the ECRs powering the experiments) when reviewer 2 seems to think that "SepF in the archaeon *Methanobrevibacter smithii* behaves largely the same as the bacterial equivalent" and "A better mechanistic insights into this peculiar SepF localization mechanism seems reasonable for a [Redacted]". I personally don't care if the paper gets published at [Redacted] or Nat Comm. That is not in my mind while serving as a scientific expert to help the paper get to the best shape possible in the least amount of time. Regardless, there are so many (subtle but not less important) mechanistic insights that set bacterial and archaeal biology apart here. The cell structure, having S-layer or pseudomurein, already poses a completely different puzzle both in the cell architecture as in the evolutionary aspect. In my opinion, this is a paper that shows that studying divergent, close to impossible to domesticate organisms

will pay off in future generations.

Reviewer #2 (Remarks to the Author):

I have reviewed the Pende et al paper for [Redacted] and my main conclusion was that the mechanistic novelties, in my opinion, were insufficient for this high-impact journal since the results more or less confirm what has been shown in earlier work with bacteria and archaea, and although there are clear differences (e.g. no polymerisation of SepF, at least in vitro), I find it difficult to appreciate these differences as being such major breakthroughs. The authors did a lot of work and established novel cell biology protocols for this archaea, but is this the same as an important mechanistic novel finding? my gut feeling says no. Now the problem is that I consider Nature Communications also a high-impact journal. Therefore, I remain of the opinion that this work does not contribute a level of novelty that really changes our view of how SepF works.

Reviewer #3 (Remarks to the Author):

This manuscript has greatly benefited from the revisions made by the authors and the careful way that most of the comments of the reviewers were addressed. As I have reviewed this manuscript previously and am satisfied with the overall modifications made I have no further remarks and hope to see this paper published shortly. I would like to congratulate the authors on a fine piece of work.

Reviewer #5 (Remarks to the Author):

The revised manuscript by Pende, Sogues et al has been largely improved by inclusion of the different comments and suggestions by the reviewers. From the structural point of view, description of the interactions between components of this unique SepF:FtsZ complex is stronger than before and provide all details required to understand mechanistic differences between bacteria and archaea cases. I also appreciate the effort in including new figures. All my previous concerns have been largely answered in this revised version and I want just felicitate the authors for the important piece of work on this very relevant topic.